# Nasal Epithelial Cells Activated with *Alternaria* and House Dust Mite Induce Not Only Th2 but Also Th1 Immune Responses

**DOI:** 10.3390/ijms21082693

**Published:** 2020-04-13

**Authors:** Seung-Heon Shin, Mi-Kyung Ye, Dong-Won Lee, Mi-Hyun Chae, Ba-Da Han

**Affiliations:** Department of Otolaryngology-Head and Neck Surgery, School of Medicine, Catholic University of Daegu, Daegu 40472, Korea; miky@cu.ac.kr (M.-K.Y.); neck@cu.ac.kr (D.-W.L.); leonen@hanmail.net (M.-H.C.); bdhan@cu.ac.kr (B.-D.H.)

**Keywords:** *Alternaria*, house dust mites, rhinosinusitis, nasal epithelial cell, innate lymphoid cell, transcription factor

## Abstract

Chronic rhinosinusitis (CRS) is a heterogeneous disease characterized by mucosal inflammation. Airborne allergens are associated with upper and lower airway inflammatory disease. We investigated the effects of airborne allergen stimulation in the nasal epithelial cells and their effect on the peripheral blood mononuclear cells’ (PBMCs) Th immune polarization. Interleukin (IL)-10, IL-25, IL-33, and thymic stromal lymphopoietin (TSLP) levels were determined using the enzyme-linked immunosorbent assay (ELISA) in nasal polyp tissues. Cultured primary nasal epithelial cells were stimulated with *Alternaria alternata, Aspergillus fumigatus*, *Dermatophagoides pteronyssinus* (DP), and *Dermatophagoides farina* (DF) for 48 hours. IL-6, IL-25, IL-33, and TSLP production were measured by ELISA, and the nuclear factor-κB (NF-κB), activator protein 1 (AP-1), and mitogen-activated protein kinase (MAPK) expression were determined by western blot analyses. PBMCs were cultured with nasal epithelial cell-conditioned media (NECM), and IL-5, interferon (IFN)-γ, and tumor necrosis factor (TNF)-α were measured. Innate lymphoid type2 cells (ILC2) were analyzed with flowcytometry. IL-25, IL-33, and TSLP levels were significantly higher in eosinophilic nasal polyps. *Alternaria*, DP, and DF enhanced IL-33 and TSLP production from the nasal epithelial cells through the NF-κB, AP-1, and MAPK pathway. NECM induced IL-5, IFN-γ, and TNF-α production from PBMCs, without increasing ILC2 expression. *Alternaria* and house dust mites enhanced the chemical mediator production from nasal epithelial cells and these allergens may induce not only Th2 inflammatory responses but also Th1 inflammatory responses in the nasal mucosa.

## 1. Introduction

Respiratory epithelial cells are the primary defense system against external pathogens with the initiation, maintenance, and regulation of innate and adaptive immune responses [1,2]. They also play an important role in the inflammatory process of the respiratory mucosa with the production of chemical mediators. Interleukin (IL)-25, IL-33, and thymic stromal lymphopoietin (TSLP) produced from nasal epithelial cells are critical regulators of Th2 immune responses at nasal mucosa [3]. IL-25 belongs to the IL-17 cytokine family produced by T lymphocytes, dendritic cells, mast cells, basophils, eosinophils and epithelial cells [4]. IL-25 supports the Th2 immune response by inhibiting interferon gamma (IFN-γ) production from the lymphocytes [5]. IL-33 is a member of the IL-1 family produced by fibroblasts, dendritic cells, endothelial cells, and epithelial cells [6]. IL-33 induces Th2 cytokine production from inflammatory cells [7]. TSLP is an IL-7 cytokine family, produced mainly by epithelial cells, epidermal keratinocytes, smooth muscle cells, fibroblasts, and dendritic cells. TSLP stimulates myeloid dendritic cells, leading to the differentiation of naïve T cells toward the Th2 cells [8].

Airborne allergens, such as house dust mites (HDM) and fungi are continuously inhaled and in contact with the nasal mucosa. Allergenic components induce the production of inflammatory chemical mediators through the interaction of cell membrane receptors, such as toll-like and protease-activated receptors. The proteases that form fungi and HDM activate and can induce the chemical mediator production from epithelial cells and promote Th2 immune responses [9,10]. *Dermatophagoides pteronyssinus* (DP) and *Dermatophagoides farina* (DF) are the major causes of the development of an allergic airway disease. *Alternaria* and *Aspergillus* are commonly found in the airway secretion and are associated with the development of sinusitis, allergic rhinitis, and bronchial asthma. Respiratory epithelial cells provide a physical barrier and produce chemokines, cytokines, and antimicrobial components to prevent or eliminate pathogenic microorganisms. Airborne allergens have been shown to trigger inflammatory mediator production from respiratory epithelial cells, and to induce airway inflammation by activating inflammatory cells independent of the acquired immunity [11,12].

Chronic rhinosinusitis (CRS) is a chronic inflammatory disease of the sinonasal mucosa with non-infectious inflammatory immune responses. It is phenotypically divided into CRS with nasal polyps (CRSwNP) and CRS without nasal polyps (CRSsNP) [13]. CRSwNP is further classified into eosinophilic and non-eosinophilic, with a poorly understood pathophysiology. Eosinophilic NP displays a significant increase in Th2 cytokines and their related mark levels.

Epithelial cell-derived IL-25, IL-33, and TSLP are significantly increased in eosinophilic CRS compared to a control [14]. Nuclear factor-κB (NF-κB), activator protein 1 (AP-1), and mitogen-activated protein kinase (MAPK) are key transcription factors associated with the induction and regulation of chemical mediators in inflammatory processes. The interaction between airborne allergen-activated nasal epithelial cells and lymphocytes, or the association of type 2 innate lymphoid cells (ILC2), are not commonly investigated. Therefore, this study aims to examine the effect of airborne allergens on IL-25, IL-33, and TSLP production, and on the expression of transcription factors from nasal epithelial cells and their effect on Th immune responses.

## 2. Results

### 2.1. Clinical Characteristics and Chemical Mediators in CRS with Nasal Polyps (CRSwNPs)

A total of 30 patients with CRSwNP were enrolled in this study: 14 with eosinophilic NP (ENP) and 16 with non-eosinophilic NP (NENP). Eight and six patients with ENP and NENP, respectively, had the allergy (*p* = 0.021). Patients with ENP commonly had olfactory dysfunction, tissue eosinophilia, and severe Lund–Mackay (LM) computed tomography (CT) score (Table 1).

The expression levels of IL-25 (6.6 ± 4.6 pg/mL), IL-33 (920.1 ± 287.7 pg/mL), and TSLP (119.7 ± 86.0 pg/mL) in patients with ENP were significantly higher than those with NENP and normal uncinate tissue (IL-25; 3.3 ± 4.2 pg/mL & 1.2 ± 1.8 pg/mL, IL-33; 709.8 ± 250.1 pg/mL & 699.7 ± 136.0 ph/mL, and TSLP; 83.0 ± 70.0 pg/mL & 40.1 ± 25.7 pg/mL). IL-10 tended to be lower in CRSwNP, but not statistically significant compared with the control group (Figure 1).

### 2.2. Production of Chemical Mediators from Nasal Epithelial Cells by Airborne Allergens

To determine the adequate stimulation time, nasal epithelial cells were cultured with 50 ug/mL and 100 ug/mL of *Alternaria alternata* for 6, 24, and 48 h. IL-33 and TSLP production significantly increased after 24 and 48 h of stimulation, but not IL-25 (Figure 2). Without stimulation, IL-33 tended to decrease over time. Then, we stimulated nasal epithelial cells with airborne allergens for 48 h for further studies.

Nasal epithelial cells were activated with 50 and 100 ug/mL of *Alternaria, Aspergillus*, DP, and DF for 48 h. IL-25 production was not increased by airborne allergens. *Alternaria* and house dust mites (DP and DF) enhanced IL-33, TSLP, and IL-6 production from nasal epithelial cells. However, *Aspergillus* only influenced the IL-6 production (Figure 3). Therefore, *Alternaria*, DP, and DF were used for further experiments. 

### 2.3. Expression of Transcription Factors from Nasal Epithelial Cells by Airborne Allergens 

We determined the expression of NF-κB, activator protein 1 (AP-1), and mitogen-activated protein kinase (MAPK) transcription factor with *Alternaria*, DP, and DF, because of these allergens’ results in cytokine IL-33 and TSLP production, which promote a Th2 immune response. *Alternaria* enhanced phosphorylated NF-κB and phosphorylated C-Jun expressions, while DP and DF enhanced NF-κB, phosphorylated NF-κB, phosphorylated C-Jun, and p38 from nasal epithelial cells. However, they did not influence extracellular signal-regulated kinase (ERK), phosphorylated ERK, jun kinase enzyme (JNK), and phosphorylated JNK expression (Figure 4). 

### 2.4. Effects of Transcription Factor Inhibitors on Chemical Mediator Production from Nasal Epithelial Cells

Because *Alternaria*, DP, and DF enhanced the IL-33 and TSLP production by increasing the NF-κB, C-Jun, and p38 expression, activation of transcription factors was blocked along with their inhibitors. *Alternaria*-induced IL-33 and TSLP productions were significantly inhibited by the NF-κB, AP-1, and MAPK inhibitors. DP-induced IL-33 was also significantly inhibited by these three inhibitors. However, DF induced IL-33 production was not suppressed by the MAPK inhibitor. DP and DF-induced TSLP productions were significantly inhibited by the AP-1 and MAPK inhibitors (Figure 5).

### 2.5. Effects of Nasal Epithelial Cell-Conditioned Media (NECM) on Peripheral Blood Mononuclear Cells (PBMCs) and Innate Lymphoid Type 2 Cell (ILC2) Expressions

Peripheral blood mononuclear cells (PBMCs) were incubated with (NECM) for 72 h. IL-5 and IFN-γ productions were significantly increased in NECM treated with *Alternaria* (IL-5, 12.6 ± 5.7 pg/mL; IFN-γ, 27.4 ± 9.5 pg/mL), DP (IL-5, 9.8 ± 4.7 pg/mL; IFN-γ, 28.36 ± 13.2 pg/mL), and DF (IL-5, 15.4 ± 7.6 pg/mL; IFN-γ, 23.9 ± 11.6 pg/mL), compared with non-treated NECM (IL-5, 5.3 ± 3.2 pg/mL; IFN-γ, 16.5 ± 10.3 pg/mL). TNF-α production was strongly increased by NECM treated with *Alternaria* (230.2 ± 96.6 pg/mL), DP (186.1 ± 106.8 pg/mL), and DF (314.7 ± 133.1 pg/mL), compared with non-treated NECM (35.2 ± 12.7 pg/mL). 

After 72 h of treatment with NECM, PBMCs were stained to identify ILC2 expression. Lymphocytes were identified from whole PBMCs, and Lin- cells were gated and further assessed for co-expression of CD127 and CRTH2. ILC2s were identified as Lin-CRTH2+CD127+ lymphocytes. The number of ILC2s was not significantly increased in PBMCs after the NECM treatment, compared with the negative control. Figure 6 shows the representative flowcytometric finding of PBMCs for ILC2. 

## 3. Discussion

CRS is phenotypically divided into CRSwNP and CRSsNP. Th2 inflammation is commonly associated with CRSwNP, whereas Th1 inflammation is associated with CRSsNP. However, CRS had a combination of Th1, Th2, and other inflammatory patterns with mixed endotypes [15]. ILC2s are increased in CRSwNP or eosinophilic CRS and epithelial cell-derived IL-25, IL-33, and TSLP are the key cytokines to induce Th2 cytokine production by ILC2s [16,17]. Although *Alternaria*, DP, and DF enhanced the IL-33 and TSLP production from nasal epithelial cells, their conditioned media induced the Th1, Th2, and pro-inflammatory cytokine production from PBMCs. 

IL-25, IL-33, and TSLP were increased in the tissue sample of CRS and these mediators were more strongly expressed in ENP, which is consistent with other studies [14,18]. TSLP was also significantly higher in NENP, compared with control tissues. Among the three cytokines, IL-33 was most strongly expressed in sinonasal mucosa. CRS and nasal polyps are developed with the predisposition of allergic airway inflammation [19]. Because airborne allergens have been known to induce the chemical mediator production from nasal epithelial cells, we stimulated nasal epithelia cells with two different fungi and house dust mites. *Alternaria*, DP, and DF induced IL-33 and TSLP production from nasal epithelial cells but not IL-25. Although IL-25, IL-33, and TSLP are expressed in the epithelial area of sinonasal tissues, previous studies have shown that IL-25 is mainly produced by Th2 cells and mast cells [20,21]. These findings suggest that airborne allergens may increase tissue IL-25 levels through the interaction of epithelial cells and other IL-25 producing inflammatory cells. 

Toll-like receptor (TLR) 1 is associated with the production of IL-25 and IL-33, while TLR2 is associated with TSLP production, and IL-33 is induced by treatment with the TLR2, TLR4, and TLR9 ligands [22,23,24]. *Alternaria* has been known to induce chemical mediator production in nasal epithelial cells through the TLR2, TLR3, or TLR4 pathways [25]. House dust mites (HDM) can activate respiratory epithelial cells through TLR 2 and TLR4 [23]. *Alternaria* and HDM can activate respiratory epithelial cells through the NF-κB, AP-1, and MAPK pathways [26,27]. In this study, *Alternaria* and HDM induced the IL-33 production and TSLP interaction through the NF-κB, AP-1, and MAPK pathways. NF-κB, AP-1, and MAPK inhibitors suppressed the production of IL-33 and TSLP from nasal epithelial cells. These transcription factor inhibitors could be used as therapeutic or preventive agents for CRS. Moreover, epithelial cell membrane TLRs may be associated with IL-33 and TSLP production from nasal epithelial cells. 

NECM produced by *Alternaria* and HDM induced IL-5, TNF-α, and INF-γ from PBMCs. In this study, IL-25, IL-33, and TSLP in NECM were produced by nasal epithelial cells and they can also produce several other chemical mediators by *Alternaria* and HDM, such as granulocyte macrophage colony stimulating factor (GM-CSF), IL-1β, IL-8, etc. [22,25]. The activation of nasal epithelial cells may induce not only Th2 inflammation but also Th1 inflammation in nasal mucosa through the interaction between structural and inflammatory cells. In CRS with or without nasal polyps, although CRSwNP has Th2 dominant and CRSsNP has Th1 dominant endotype characteristics; approximately 30% of CRS had mixed endotypes [15]. In the nasal mucosa, which contains epithelial cells, structural cells, and other inflammatory cells, exogenous pathogenic stimulation simultaneously triggers not only epithelial cells but also structural cells and the produced chemical mediators may have autocrine or paracrine effects, which enhance the inflammatory reaction in the nasal mucosa. Because *Alternaria* and HDM enhanced IL-33 and TSLP production in nasal epithelial cells, NECM is expected to increase Th2 cytokines and ILC2 in PBMCs. However, NECM induced Th1 and Th2 cytokine production, but did not influence ILC2 expression. Although *Alternaria* and HDM induced IL-33 and TSLP production from nasal epithelial cells, the concentration of these two chemical mediators in cultured media was much lower than that of nasal polyp tissues. The cellular proportion of ILC2 in PBMCs is very low and was less than 0.1% in patients with allergic rhinitis. The concentration of epithelial cell-derived chemical mediators can insufficiently polarize PBMCs to ILC2, or the polarized ILC2 number might not be enough to detect with the flowcytometric technique. We used two different allergens, fungi and HDM. Although these allergens can activate nasal epithelial cells, antigens contain several different protein and enzymatic allergens, which interact with nasal epithelial cells through various immune receptors, such as TLR, protease activated receptors, G-protein-coupled receptors, etc., and induce the production of chemical mediators through different transcription pathways. Although IL-33 and TSLP productions were enhanced by *Alternaria* and HDM, their exact immunopathologic mechanism may be different. In this study, we used normal nasal epithelial cells, which could not represent the CRS or nasal polyp epithelial cells. If we used pathologic mucosal epithelial cells, they might show different or stronger immune responses against airborne allergens. 

## 4. Materials and Methods 

### 4.1. Chemical Mediators in CRSwNP

The chemical mediators were evaluated from 38 subjects. A total of 30 nasal polyps were obtained from 14 patients with eosinophilic NP (ENP) and 16 with non-eosinophilic NP (NENP). Eight uncinate process mucosa were used as the control tissue, which was obtained from patients with a medial blowout fracture or endoscopic dacryocystostomy. CRSwNP was confirmed by computed tomography and endoscopic examinations in accordance with the “European position paper on rhinosinusitis and nasal polyps 2020” guidelines [19]. All patients were aged above 18 years. Patients were excluded if they had received systemic or topical steroids, and had taken antibiotics or antihistamines during the four weeks that preceded the study. This study was approved by the institutional review board of Daegu Catholic University Medical Center, and all patients signed a consent form that outlined the objectives of the study (CR-17-152, 13 December 2017).

Freshly obtained tissue specimens were homogenized and centrifuged at 3000 rpm for 10 minutes at 4 °C. After centrifugation, 500 μL aliquots of the supernatant were assayed for IL-10, IL-25, IL-33, and TSLP using commercially available enzyme linked immunosorbent assay (ELISA) kits (R&D Systems, Minneapolis, MN, USA). The chemical mediator data representing protein concentrations was calculated based on 40 μg of protein.

### 4.2. Stimulation of Primary Nasal Epithelial Cells with Airborne Allergens

Primary nasal epithelial cells were isolated from the inferior turbinate of 10 subjects (6 men and 4 women; 45.2 ± 11.8 years) during the septal surgery. Subjects were excluded if they had an active inflammation, allergy, and had received antibiotics, antihistamine, or other medications for at least four weeks preoperatively. Allergy status was determined using the skin prick test.

Specimens were placed in Ham’s F-12 medium supplemented with 100 IU penicillin, 100 ug/mL streptomycin, and 2 ug/mL amphotericin B. Nasal mucosa was rinsed with Ham’s F-12 medium supplemented with antibiotics and incubated with 0.1% dispase (Roche Diagnostics, Mannheim, Germany) for 16 hours at 4 °C. The epithelial cells were isolated by gentle agitation and cell suspensions were filtered through a No. 60 mesh cell dissociation sieve. The cells were suspended in Ham’s F-12 medium supplemented with antibiotics, 150 ug/mL glutamine, 5 ug/mL transferring, 25 ng/mL epithelial growth factor, 15 ug/mL endothelial cell growth supplement, insulin 5 IU/mL 200 pM triiodothyronin, 100 nM hydrocortisone, and 15% fetal calf serum (FCS). Cell suspensions (10^6^ cells/mL) were plated in culture plates and placed in a 5% CO_2_ humidified incubator at 37 °C. Epithelial cell cultures that reached 80%–90% confluence were treated with 100 ug/mL and 50 ug/mL of culture filtrated *Alternaria alternata* and *Aspergillum fumigatus,* or crushed DP and DF (Greer Lab, Lenoir, NC, USA). After 48 h of incubation, supernatants and epithelial cells were harvested and stored at −70 °C until assayed. IL-25, IL-33, TSLP, and IL-6 were quantified using the ELISA kit (R&D system, Minneapolis, MN, USA). 

### 4.3. Western Blot Analysis of Nasal Epithelial Cells for Transcription Factors

After 1 hour of treatment with airborne allergens, nasal epithelial cells were harvested and lysed in an ice-cold lysis buffer (Thermo Scientific, Rockford, IL, USA). Collected nasal epithelial cell lysates were subjected to sodium dodecyl sulfate polyacrylamide gel electrophoresis and transferred onto NC membranes (Bio-Rad, Berkeley, CA, USA). Membranes were blocked with 5% skim milk solution and incubated with antibodies against nuclear factor κB (NF-κB), phosphorylated NF-κB, C-Jun, phosphorylated C-Jun, p38, phosphorylated p38, ERK, phosphorylated ERK, JNK, phosphorylated JNK, and Glyceraldehyde-3-phosphate dehydrogenase (GAPDH) (Santa Cruz Biotechnology, Santa Cruz, CA, USA). After a 1-hour incubation, the membranes were washed and then treated with peroxidase-conjugated anti-rabbit immunoglobulin G (Santa Cruz Biotechnology). Bands were visualized using horseradish peroxidase-conjugated secondary antibodies and an enhanced chemiluminescence (ECL) system (Pierce, Rockford, IL, USA). Band densities were measured using the Multi Gauge v.2.02 software (Fujifilm, Tokyo, Japan) and expressed as a percentage of treated vs. untreated cells.

### 4.4. Effects of Transcription Factor Inhibitors on Cytokine Production

Nasal epithelial cells were pretreated with BAY 11-7082 as a NF-κB inhibitor, curcumin as an activator protein 1 (AP-1), and SB 203580 as a mitogen-activated protein kinase (MAPK) inhibitor (Calbiochem, San Diego, CA, USA). After a 1-hour treatment, epithelial cells were stimulated with airborne allergens for 48 hours, and then the supernatant was collected to determine the IL-33 and TSLP levels.

### 4.5. Effects of Nasal Epithelial Cell Conditioned Media (NECM) on Peripheral Blood Mononuclear Cells (PBMCs) and Innate Lymphoid Type 2 Cell (ILC2s) Expressions

PMBCs were isolated from heparinized blood by the density gradient centrifugation method with Histopaque (Sigma, St Louis, MO, USA). 2 × 10^6^ cells were cultured with stimulated NECM for 72 h, and then IL-5, IFN-γ, and tumor necrosis factor (TNF)-α production was measured with an ELISA kit (R&D system). To determine ILC2 expression with flowcytometry, cells were stained as described previously [26]. PBMCs were stained with a FITC CD2, CD3, CD14, CD16, CD19, CD56, CD235 lineage cocktail (eBioscience, San Diego, CA, USA) and FcεRI. Moreover, they were further stained with phycoerythrin (PE)-conjugated chemoattractant receptor–homologous molecule expressed on Th2 lymphocytes (CRTH2) and PE-Cy7 conjugated CD127 (BD Biosciences, Franklin Lakes, NJ, USA), and fixed with 4% paraformaldehyde. ILC2s were identified as Lin-CRTH2+CD127+ lymphocytes with a flowcytometer machine (Beckman Coulter, Hercules, CA, USA). 

### 4.6. Statistical Analysis

All measured parameters are expressed as mean ± standard deviation. Clinicopathological differences between eosinophilic and non-eosinophilic CRS were determined using two sample *t*-tests and a chi-squared test. A one-way analysis of variance followed by a Tukey’s test was performed for normally distributed data, and the Kruskal–Wallis test with post-hoc Bonferroni–Dunn test was performed for non-normally distributed data. The analysis was conducted with the Statistical Package of Social Sciences software version 21 (SPSS Inc., Chicago, IL, USA). Results with *p* < 0.05 were regarded as statistically significant.

## 5. Conclusions

The results of this study demonstrated that *Alternaria*, and HDM induced IL-33 and TSLP production from nasal epithelial cells through the NF-κB, AP-1, and MAPK pathways. *Alternaria*-induced IL-33 and TSLP productions were significantly suppressed by NF-κB, AP-1, and MAPK inhibitors, however HDM-induced TSLP production was not suppressed by the NF-κB inhibitor. These findings suggest that *Alternaria* and HDM may induce the production of chemical mediators in a different immune pathway. 

NECM produced by nasal epithelial cells activated with *Alternaria* and HDM enhanced the Th1, Th2, and pro-inflammatory cytokine production from PBMCs. Although we focused on IL-33 and TSLP production from nasal epithelial cells, many other chemical mediators could be produced by *Alternaria* and HDM. These finding suggest that although *Alternaria* and HDM are commonly known for being part of a Th2 inflammation associated with an allergen, these allergens could induce Th2 and other inflammation responses in the nasal mucosa.

## Figures and Tables

**Figure 1 ijms-21-02693-f001:**
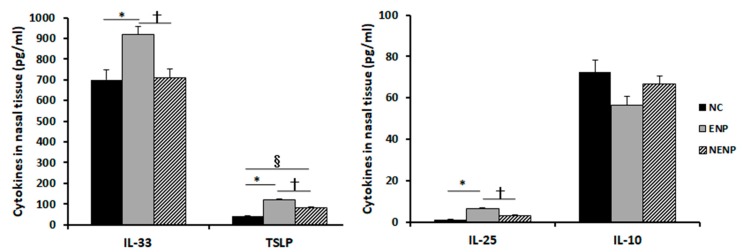
Chemical mediator levels in the nasal polyps and normal uncinate process. Interleukin (IL)-25, IL-33, and thymic stromal lymphopoietin (TSLP) expressions were significantly higher in the eosinophilic nasal polyps than other groups. Negative control (NC); eosinophilic nasal polyp (ENP), non-eosinophilic nasal polyp (NENP); * *p* < 0.05 between NC vs. ENP; † *p* < 0.05 between ENP vs. NENP; § *p* < 0.05 between NC vs. NENP.

**Figure 2 ijms-21-02693-f002:**
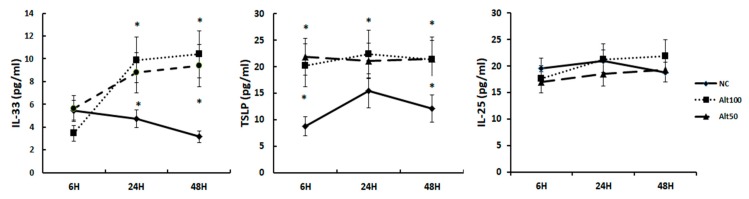
Kinetic studies to determine the optimal stimulation interval. IL-33, TSLP, and IL-25 production were significantly increased after 24 and 48 h stimulation of the nasal epithelial cells with 50 and 100 ug/mL of *Alternaria*. Negative control (NC); 100 ug/mL of *Alternaria* (Alt100); 50 ug/mL of *Alternaria* (Alt50); * *p* < 0.05 compared with NC.

**Figure 3 ijms-21-02693-f003:**
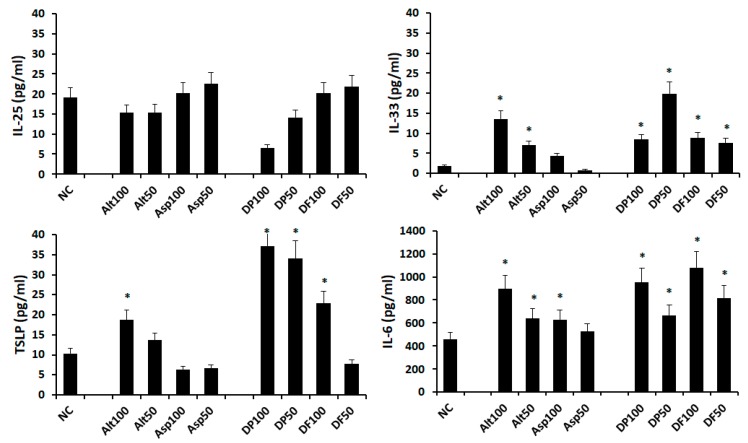
Production of chemical mediators by airborne allergens from nasal epithelial cells. *Alternaria* (Alt), *Dermatophagoides pteronyssinus* (DP), and *Dermatophagoides farina* (DF) significantly enhanced the IL-33, TSLP, and IL-6 production. Negative control (NC); *Aspergillus* (Asp), 50; 50 ug/mL, 100; 100 ug/mL. * *p* < 0.05 compared with NC.

**Figure 4 ijms-21-02693-f004:**
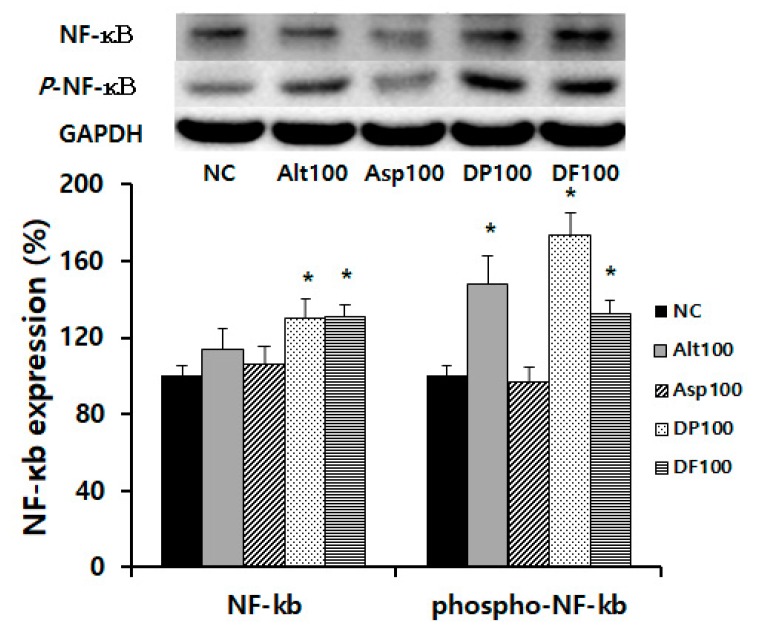
Expression of transcription factors by airborne allergens from nasal epithelial cells. *Alternaria* (Alt), *Dermatophagoides pteronyssinus* (DP), and *Dermatophagoides farina* (DF) significantly enhanced the phosphorylated nuclear factor-κB (NF-κB), phosphorylated C-Jun, and phosphorylated p38 expressions. DP and DF significantly enhanced NF-κB and p38 expression. Negative control (NC); *Aspergillus* (Asp), 100; 100 ug/mL. * *p* < 0.05 compared with NC.

**Figure 5 ijms-21-02693-f005:**
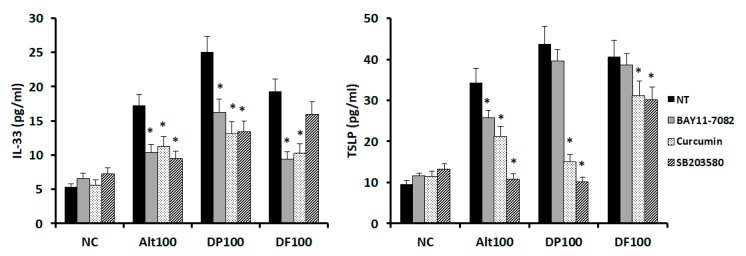
Effects of transcription factor inhibitors on the IL-33 and TSLP production from nasal epithelial cells. *Alternaria* induced IL-33 and TSLP production were significantly inhibited by BAY 11-8082 (NF-κB inhibitor), curcumin (activator protein 1 (AP-1) inhibitor), and SB 203580 (mitogen-activated protein kinase (MAPK) inhibitor). DP and DF-induced IL-33 and TSLP production was also suppressed with transcription factor inhibitors. Negative control (NC); not treated (NT), 100; 100 ug/mL, * *p* < 0.05 compared with NT.

**Figure 6 ijms-21-02693-f006:**
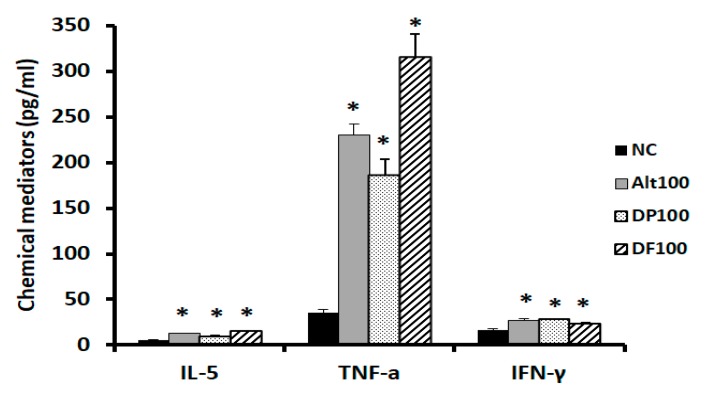
Effects of nasal epithelial cell conditioned media (NECM) on the production of chemical mediators in peripheral blood mononuclear cells. TNF-α and INF-γ productions were significantly increased by NECM treated with *Alternaria*, *Dermatophagoides pteronyssinus* (DP), and *Dermatophagoides farina* (DF). Negative control (NC), nasal epithelial cell-conditioned media (NECM) treated with 100 ug/mL of *Alternaria* (Alt100), NECM treated with 100 ug/mL of DP (DP100), NECM treated with 100 ug/mL of DF (DF100), * *p* < 0.05 compared with NC.

**Table 1 ijms-21-02693-t001:** Demographic characteristics of eosinophilic and non-eosinophilic nasal polyps.

	ENP (*n* = 14)	NENP (*n* = 16)	*p* Value
Atopy	8 (57.1%)	6 (37.5%)	0.021
PB eosinophilia (>5%)	7 (50.0%)	2 (12.5%)	<0.001
OFT			<0.001
Normosmia	2	5	
Hyposmia	3	7	
Anosmia	9	4	
Bilaterality	13 (92.9%)	11 (68.8%)	0.026
LM score	7.9 ± 2.3	7.2 ± 2.4	0.137

Eosinophilic nasal polyp (ENP), non-eosinophilic nasal polyp (NENP), peripheral blood (PB), olfactory function test (OFT), Lund–Mackay (LM).

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
