# Peer review of "Nasal Epithelial Cells Activated with Alternaria and House Dust Mite Induce Not Only Th2 but Also Th1 Immune Responses"

_ijms, 2020, doi:10.3390/ijms21082693_

Round 1

Reviewer 1 Report

The authors proposed an interesting study on the complex pathways that drive the nasal mucosa inflammatory response to allergens, particularly focusing on the activation of Th1 inflammatory response. The study presents an accurate methodology and the results obtained are well discussed.

I would signal to the authors some minor issues.

  1.   Introduction: the authors well resume fundamental information regarding nasal epithelium role in inflammatory response. I believe the “physiological à pathological” passage (from inflammatory response to CRS) at line 60 results too much sharp. I suggest adding a logical step.
  2.   Discussion: lines 171-172.  I suppose the two sentences are separated by mistake.
  3.   Discussion / Conclusion: I’ve found fascinating that the authors tested the effect of NF-κB, AP-1, and MAPK inhibitors.   I would like the authors to comment on how practical and the utility of this knowledge is - is this something we could use as a therapy for CRS in the next future? 

Author Response

I thank the editors and referees of the ‘International Journal of Molecular Sciences’ for taking their time to review my article. I have made some corrections and added some results in the manuscript after going over the referee’s comments.

  1.   Introduction: the authors well resume fundamental information regarding nasal epithelium role in inflammatory response. I believe the “physiological à pathological” passage (from inflammatory response to CRS) at line 60 results too much sharp. I suggest adding a logical step.

    Answer) To mention about innate immune system of epithelial cells ‘Respiratory epithelial cells provide physical barrier and produce chemokines, cytokines, and antimicrobial components to prevent or eliminate pathogenic microorganisms.’ was added at line 60.

  2. Discussion: lines 171-172.  I suppose the two sentences are separated by mistake.

    Answer) Two sentences were added as one.
  3.   Discussion / Conclusion: I’ve found fascinating that the authors tested the effect of NF-κB, AP-1, and MAPK inhibitors.   I would like the authors to comment on how practical and the utility of this knowledge is - is this something we could use as a therapy for CRS in the next future?  

    Answer)
    NF-κB, AP-1, and MAPK inhibitors could inhibit the production of IL-33 and TSLP from nasal epithelial cells. But Alternaria and HDM may have different immune pathway to produce chemical mediators. Transcription factor inhibitors had some effects. So we added ‘NF-κB, AP-1, and MAPK inhibitors suppressed the production of IL-33 and TSLP production ……………… as a therapeutic or preventive agent for CRS.‘ in Discussion line 187.

I hope the revised manuscript will better meet the requirements of the ‘International Journal of Molecular Sciences’ for publication.

Thank you.

Reviewer 2 Report

In this research article, authors used ELISA assays to quantify the production of IL-25, IL-33, TSLP and other inflammatory factors by cultured primary nasal epithelial cells stimulated with different allergens (house dust mite, Alternaria). Western blot were used to describe NF-κB, AP-1 and MAPK expression/activation. Results shown that both house dust mite and Alternaria induce TH2 and TH1 signals by nasal epithelial cells. The manuscript could be improved and some control seems missed. This manuscript could be suitable for publication in IJMS after major revision.

Major points:

  • Figure 2: how do you explain the decrease of IL-33 in control group over the time.
  • Figure 4: could you use alternative methods to show activation of these actors. Such as imagery to show NF-κB nuclear translocation?
  • In figure 3, Aspergillus do not induces production of TSLP, IL-25, IL-33 and IL-6. It could be add as negative control in your experiments for figure 4 results.
  • Line 123: authors used transcription factor inhibitors. It could be important to prove the efficacy of these inhibitors in presented experimental approaches.
  • Figure 6: which sample is presented in the flow cytometry figure? Authors should provide raw data for all tested groups. Moreover, a positive control is needed to show that the gating strategy is appropriate to detect ILC2.

Minor points:

  • Figure 1: could you ameliorate the design of your figure in order to see better the amount and the differences between each group for IL-25 test?
  • Figure 2: show IL-25 results, even if increase production is not observed. Is a decrease observed?
  • Line 109: it could be interesting to present NF-κB, AP-1 and MAPK and their role in the introduction.
  • Line 113: add a dot after “epithelial cell”
  • Figure 6: replace IFN-r by IFN-γ
  • Could you provide more informations about NECM medium (origin, composition)?
  • Could you provide more informations about the allergens (house dust mite, Alternaria) used (inactivated samples, extract, total extract, presence of endotoxin).

Author Response

I thank the editors and referees of the ‘International Journal of Molecular Sciences’ for taking their time to review my article. I have made some corrections and added some results in the manuscript after going over the referee’s comments.

Major points:

  • Figure 2: how do you explain the decrease of IL-33 in control group over the time.

    Answer) Although we cannot explain exactly, the production of IL-33 decreases over time. It might be associated with the limitation of intracellular amount of IL-33 level. To clarify ‘Without stimulation, IL-33 tended to decrease over time.’ was added in line 93.

  • Figure 4: could you use alternative methods to show activation of these actors. Such as imagery to show NF-κB nuclear translocation?  
  •  
  • Answer) In figure 4, Aspergillus 100 results were added in figure 4.
    We did not change figure 4, if we changed with imagery finding, it would be very complicate to understand.
  • In figure 3, Aspergillus do not induces production of TSLP, IL-25, IL-33 and IL-6. It could be add as negative control in your experiments for figure 4 results.
  • Line 123: authors used transcription factor inhibitors. It could be important to prove the efficacy of these inhibitors in presented experimental approaches.

  • Answer) It was mention in Discussion as ‘NF-κB, AP-1, and MAPK inhibitors suppressed the production of IL-33 and TSLP production from nasal epithelial cells. These transcription factor inhibitors could be used as therapeutic or preventive agent for CRS.’ at line 187.
  • Figure 6: which sample is presented in the flow cytometry figure? Authors should provide raw data for all tested groups. Moreover, a positive control is needed to show that the gating strategy is appropriate to detect ILC2.

    Answer) We measured ILC2s in PBMC with flowcytometery. The ILC2 proportion is very low in peripheral blood and hard to detect. We also could not find the effect of NECM on ILC2 expression. In Fig 6, flowcytometric finding was deleted to reduce the confusion for readers.

    Minor points:

  • Figure 1: could you ameliorate the design of your figure in order to see better the amount and the differences between each group for IL-25 test?

    Answer) To improve the quality of figure 1, it was divided into two graphs.  

  • Figure 2: show IL-25 results, even if increase production is not observed. Is a decrease observed?

    Answer) IL-25 result was added in figure 2.

  • Line 109: it could be interesting to present NF-κB, AP-1 and MAPK and their role in the introduction.

    Answer) NF-κB, AP-1 and MAPK were mentioned in the Introduction as ‘Nuclear factor-κB (NF-κB), activator protein 1 (AP-1), and mitogen-activated protein kinase (MAPK) are key transcription factors associated with induction and regulation of chemical mediators in inflammatory process. In line 66.

    Line 113: add a dot after “epithelial cell”

    Answer) Corrected as recommended. 

  • Figure 6: replace IFN-r by IFN-γ

    Answer) Corrected as recommended.

  • Could you provide more informations about NECM medium (origin, composition)?

    Answer) Nasal epithelial cell conditioned media (NECM) is supernatants, which were collected after stimulation with Alternaria or HDM. NECM contains several chemical mediators produced by nasal epithelial cells. Many papers used epithelial cell conditioned media for basic or clinical studies.
  • Could you provide more informations about the allergens (house dust mite, Alternaria) used (inactivated samples, extract, total extract, presence of endotoxin).

    Answer) Culture filtrated fungi and crushed HDM were used in this study, which were supplied from Greer Lab. The endotoxin activities of products were less than 11 EU/mg extract, which did not induce airway sensitization or inflammation. To clarify the characteristics of allergens ‘culture filtrated ….. or crushed ….’ was added in line 257.

    I hope the revised manuscript will better meet the requirements of the ‘International Journal of Molecular Sciences’ for publication.

    Thank you.

Round 2

Reviewer 2 Report

The authors addressed my concerns. The manuscript is suitable for publication.